# Bidirectional Associations between Physical Activity and Sleep in Early-Elementary-Age Latino Children with Obesity

**DOI:** 10.3390/sports9020026

**Published:** 2021-02-11

**Authors:** Justin J. Merrigan, Kristina M. Volgenau, Allison McKay, Robyn Mehlenbeck, Margaret T. Jones, Sina Gallo

**Affiliations:** 1School of Kinesiology, George Mason University, Manassas, VA 20110, USA; jmerrig2@gmu.edu (J.J.M.); amckay4@gmu.edu (A.M.); mjones15@gmu.edu (M.T.J.); 2Rockefeller Neuroscience Institute, West Virginia University, Morgantown, WV 26505, USA; 3Department of Psychology, George Mason University, Fairfax, VA 22030, USA; kvolgena@gmu.edu (K.M.V.); rmehlenb@gmu.edu (R.M.); 4School of Sport, Recreation and Tourism Management, George Mason University, Fairfax, VA 22030, USA; 5Department of Foods and Nutrition, University of Georgia, Athens, GA 30602, USA

**Keywords:** childhood, lifestyle factors, low-income, health disparities, exercise, youth, body mass index

## Abstract

Low-income Latino children are at high risk for obesity and associated comorbidities. Considering the health benefits of proper sleep habits and physical activity, understanding the patterns, or the relationship between these modifiable factors may help guide intervention strategies to improve overall health in this population. Thus, the purpose was to investigate bidirectional associations between physical activity and sleep among Latino children who are overweight/obese. Twenty-three children (boys, 70%; overweight, 17%; obese, 83%) (age 7.9 ± 1.4 years) wore activity monitors on their wrist for 6 consecutive days (comprising 138 total observations). Hierarchical linear modeling evaluated temporal associations between physical activity (light physical activity, LPA; moderate to vigorous activity, MVPA) and sleep (duration and efficiency). Although there was no association between MVPA and sleep (*p* > 0.05), daytime LPA was negatively associated with sleep duration that night (estimate ± SE = −10.77 ± 5.26; *p* = 0.04), and nighttime sleep efficiency was positively associated with LPA the next day (estimate ± SE = 13.29 ± 6.16; *p* = 0.03). In conclusion, increased LPA may decrease sleep duration that night, but increasing sleep efficiency may increase LPA the following day. Although further investigation is required, these results suggest that improving sleep efficiency may increase the level of physical activity reached among Latino children who are overweight/obese.

## 1. Introduction

Factors that influence the prevalence of obesity in children, include culture, environment, behaviors, and low socioeconomic status (SES), with some more difficult to alter than others [1]. Latino children have some of the highest rates of obesity in the United States with disparities apparent early in life or in preschool years [2]. Modifiable factors, such as physical activity level, can be more easily addressed. Increased time in sedentary activities leads to a greater likelihood of obesity, cardiovascular disease, and metabolic syndrome [3,4,5]. However, younger children (6–11 years) appear to exceed physical activity guidelines of at least 60 min of moderate to vigorous physical activity (MVPA), regardless of ethnicity and weight [3]. This is also true for Latino children, who perform more MVPA than their white counterparts [3]. Yet, children who are overweight/obese may have more difficulty with weight bearing tasks and thus, have difficulty meeting MVPA guidelines [6]. It is unknown how Latino children, who are overweight/obese, perform, and considering the important role of physical activity in decreasing obesity, it is critical to investigate the physical activity patterns in this high-risk group.

During the early stages of development, adequate sleep is critical to overall health and maturation [7]. Nightly sleep duration ranging from 9–12 h has been suggested for children aged 6–12 years [8]. Children who achieve these recommended levels of sleep are more physically active [9] and have lower rates of obesity [10]. Conversely, there is evidence that those who participate in regular exercise experience improved sleep quantity (i.e., duration) and quality (i.e., sleep efficiency [11]) [12]. Recent research with children and adolescents has noted that the relationships between sleep duration and efficiency with physical activity is bidirectional [13,14,15]. A decrease in sleep duration and efficiency has been noted as a result of longer physical activity durations the prior day, while longer sleep durations and sleep efficiency reduced the next day’s physical activity [14]. However, other researchers found longer MVPA to increase that night’s sleep duration [13,15] and efficiency [13], while longer and more efficient sleep resulted in longer durations of MVPA the next day [13,15]. In a group of Spanish children, increased light physical activity (LPA) and MVPA were associated with decreased sleep duration but increased sleep efficiency, while sleep duration and efficiency were negatively and positively related with the next day’s LPA only [16]. The contradictory findings and lack of research examining these relationships in young, obese, and at-risk populations warrants further research describing these relationships.

Furthermore, Latino children from low SES experience more sleep problems, potentially due to later bedtimes combined with early wake-time requirements [17]. In order to inform interventions and policies that improve the health of this underserved population, it is of interest to investigate the bidirectional associations of sleep and physical activity. Therefore, this study aims to investigate the bidirectional associations among physical activity and sleep patterns in overweight/obese Latino children of low SES.

## 2. Materials and Methods

### 2.1. Design and Participants

This secondary cross-sectional analysis utilizes data from children who participated in a group-based pediatric weight management program (Vidas Activas y Familias Saludables) [18] and wore activity monitors for 6 days for a total of 138 observations. In brief, this was a pre-test–post-test pilot study among children and their families from Northern Virginia who participated in a 10-week community program which addressed diet, exercise, and behavior modification. The program was evidence-based [19,20] and culturally adapted by the research team and a community advisory board to integrate key sociocultural characteristics of the population. Participants (*n* = 23) were recruited from health clinics, elementary schools, and community centers that serve low-income and Latino populations. Inclusion criteria included the following: self-reported Latino (confirmed by parental country of birth or language spoken at home), body mass index (BMI-for-age/sex ≥ 85th percentile) based on CDC growth charts [21], no prior medical or behavioral diagnoses, and no regular medications. Parents provided informed consent and children’s assent prior to participation. George Mason University’s Institutional Review Board for Human Subjects approved the study and all procedures (#1094469).

### 2.2. Measures

#### 2.2.1. Physical Activity

Physical activity measures were collected over 6 consecutive days with a wrist-worn inertial measurement unit located on the non-dominant arm (ActiGraph GT9X Link, Actigraph, Pensacola, FL, USA) at a rate of 100 Hz. The data from the inertial measurement unit were downloaded in Actilife software (v6.13.4, Actigraph, Pensacola, FL, USA) and summed over 60 s epochs [22]. Hospital bands were utilized to reduce the likelihood of participants removing the device from their wrist, but some non-wear time periods were determined using previously validated methods [23]. Participants that removed their device for extended periods (i.e., >5 h) were removed from analysis (*n* = 5). At least 4 valid weekdays and 2 valid weekend days were required to be included in the analysis. Using the Actilife software, daily activity was categorized, by minutes, into groups using the following previously validated cut points: sedentary, 0–99; light, 100–2240; moderate, 2241–3840; and vigorous, >3841 [22]. For the purposes of this analysis, light physical activity (LPA; e.g., household tasks, playing catch) and MVPA (e.g., running, team sports) were included. MVPA was the sum of moderate and vigorous activities.

#### 2.2.2. Sleep

Sleep measures used the same wrist-worn inertial measurement unit (ActiGraph GT9X Link, Actigraph, Pensacola, FL, USA) for 6 consecutive days. A trained researcher visually confirmed sleep periods from the downloaded data, which were calculated using a validated algorithm [24]. Specifically, a decrease in activity levels suggested the start of bedtime, while a sudden rise in physical activity indicated wake time. Sleep periods were split into two intervals if an awakening occurred for more than one hour. Duration of sleep was calculated as the number of minutes between sleep onset and offset overnight. If two intervals occurred overnight, the sum of sleep duration for both intervals was used for analysis. Maintenance efficiency of sleep was considered sleep quality and defined as a percentage of the minutes of actual sleep divided by the duration of sleep.

### 2.3. Statistical Analysis

Due to the three-level structure of our data, interclass coefficients (ICCs) were used to examine the interdependence of outcome variables. According to the ICCs, none of the outcome variables varied across days, though they varied significantly across individuals. For this reason, a series of bidirectional two-level models were conducted to examine relationships between sleep (i.e., sleep duration and efficiency) and physical activity (i.e., LPA and MVPA) nested within individuals. All predictors were grand-mean centered prior to analysis. Through the use of nested model comparison tests, the best fitting models were selected. Analyses were conducted using maximum likelihood estimation in R 3.6.3 [25]. with alpha level of (*p* = 0.05).

## 3. Results

Overall, 23 participants completed 6 observations of sleep and physical activity (total *n* = 138). Physical characteristics, as well as physical activity and sleep data, are provided in Table 1. Data collection procedures for anthropometrics and body composition are described elsewhere [6]. Results of the hierarchical linear modeling indicated that daytime LPA was a significant predictor of that night’s sleep duration, while nighttime sleep efficiency was a significant predictor of the next day’s LPA (Table 2). For every single unit increase in daytime LPA, there was a −10.77 ± 5.26 min decrease in sleep duration that night. However, there was a 13.29 ± 6.16 min increase in the following day’s LPA for every single unit increase in sleep efficiency the prior night. No bidirectional relationships existed between MVPA and sleep (Table 2).

## 4. Discussion

This study investigated bidirectional associations between physical activity and sleep patterns in young Latino children of low SES who were overweight or obese. A bidirectional relationship between LPA and sleep duration and efficiency was not observed. However, those with above-average LPA had a shorter sleep duration that night, in agreement with prior literature investigating sleep and physical activity in Spanish children [16]. Contrary to prior evidence, participation in regular exercise, regardless of intensity, yields benefits on sleep duration and quality that night [12]. However, in 9 to 11 year old multinational children, MVPA predicted decreased sleep duration, while LPA the day prior did not predict sleep duration [15]. In the same study, longer sleep durations in the preceding night predicted increased LPA [15]. Nevertheless, other research suggests that children with shorter sleep durations are more physically active during the day [14]. One possible explanation is the inherently greater physical activity levels of children, during the day and night, that would reduce the amount of time left in the day to sleep. Thus, moving more leaves less time for quality nighttime sleep. It is important to remember the importance of health around the 24-h movement behavior guidelines, and the health impacts of each behavior without focusing all attention on one behavior [26]. It should also be mentioned that the children in this study exceeded the daily physical activity recommendations but fell below the recommended nightly sleep [9]. Those that slept with above-average efficiency, spent more time performing LPA the next day. These findings are in agreement with prior findings of Spanish children participating in more light physical activity following nights consisting of greater sleep efficiency [16]. Thus, although sleep duration was below recommended levels, improving sleep quality may help to improve activity levels.

Contrary to our hypothesis, a bidirectional relationship did not exist between MVPA and sleep measures. In previous works studying children (mean age = 15), every additional hour of MVPA above the group average increased sleep duration and improved sleep efficiency [13]. Additionally, greater amounts of MVPA and lower sedentary behavior is associated with greater next night’s sleep duration in children aged 9–11 [15]. This would suggest that MVPA is beneficial for promoting healthier sleep patterns. However, these children were older than those in the current study, and younger children seem to exceed the physical activity guidelines more than older children [27]. As such, MVPA in prior studies were 45–60 min, while children in the current study reached MVPA of approximately 3 h a day. This not only exceeds older recommendations but meets newer recommendations by the World Health Organization that children should be active for at least 3 h a day [28]. A possible explanation for this high number of MVPA is that Latino children may perform more MVPA than their white counterparts [3]. This could create a ceiling effect, where sleep time may not influence their innately high MVPA. In the other direction, some studies found that longer sleep durations and higher sleep efficiency either decreased or did not alter next day’s MVPA [13,14]. Others noted longer sleep durations the night before increased the next day’s MVPA [15]. Thus, unlike prior research, a bidirectional association between MVPA and sleep did not exist for the Latino children who were overweight/obese in the current study.

Of note, the results from this study are compiled from a small data set and should be considered preliminary findings. Although the total observations in this study provide an adequate sample size, future studies with a less homogenous sample may provide further evidence of the relationship between physical activity and sleep. Therefore, future research should consider these relationships in a variety of ethnicities, age groups, and physical activity levels. These preliminary findings are beneficial to understanding these relationships within the current sample of young Latino children with obesity, but comparisons to other populations within the same analysis are warranted. It should also be mentioned that accelerometer cut-points may contribute to the calculated physical activity and sleep for each individual. Thus, the cut-points and algorithms used to calculate physical activity and sleep patterns may influence the differences in findings among the literature.

## 5. Conclusions

The study information provides valuable insight into the physical activity and sleep patterns of an understudied population. A bidirectional relationship did not exist between LPA and sleep duration or efficiency among a sample of young Latino children who are considered overweight or obese. However, children who participate in LPA above the group’s average had a shorter sleep duration that night. Meanwhile, children that slept with above-average efficiency, spent more time performing LPA the next day. In this sample of children who exceed the current MVPA recommendations, there were no bidirectional associations between MVPA and sleep, but the majority of children fell below the recommended amount of nightly sleep. However, of note, temporal associations are not causal relationships. Further research is necessary to design and implement interventions focused on improving the health and maturation of this underserved population.

## Figures and Tables

**Table 1 sports-09-00026-t001:** Descriptive characteristics and outcomes for children participants (*n* = 23).

Variable	Mean (SD)
Age (years)	7.91 (1.35)
Sex, boys N (%)	16 (70)
Weight (kg)	44.68 (12.46)
Weight Z-score	4.6 (1.3)
Height (cm)	133.86 (8.14)
Height Z-score	1.3 (1.0)
Body mass index, BMI (kg∙m^2^)	24.51 (4.22)
BMI Z-score	2.1 (0.5)
BMI percentile (%)	97.06 (3.99)
Steps (#)	16,209 (3745)
Light physical activity (minutes)	661.05 (81.39)
Moderate to vigorous physical activity (MVPA, minutes)	186.89 (75.18)
Nightly sleep time (minutes)	451.62 (60.12)
Nightly sleep efficiency (sleep/total sleep duration, %)	84.19 (6.72)

**Table 2 sports-09-00026-t002:** Multilevel model results

**LPA Predicting Sleep Time**				**LPA Predicting Sleep Efficiency**			
Intercept	451.62	6.13	73.64 *	Intercept	84.19	4.91	101.54 *
LPA	−10.77	5.26	−2.05 *	LPA	−0.27	0.60	−0.45
**MVPA Predicting Sleep Time**				**MVPA Predicting Sleep Efficiency**			
Intercept	--	--	--	Intercept	84.19	0.84	99.72 *
MVPA	--	--	--	MVPA	9.57	0.59	0.85
**Sleep Time Predicting LPA**				**Sleep Time Predicting MVPA**			
Intercept	--	--	--	Intercept	--	--	--
Sleep Time	--	--	--	Sleep Time	--	--	--
**Sleep Efficiency Predicting LPA**				**Sleep Efficiency Predicting MVPA**			
Intercept	655.99	9.47	69.239 *	Intercept	--	--	--
Sleep Efficiency	13.29	6.16	2.157 *	Sleep Efficiency	--	--	--

Results were not reported (--) for models that were not significantly different (*p* > 0.05) from the unconditional model. Abbreviations: LPA, light physical activity; MVPA, moderate to vigorous physical activity. Results are presented as B estimate ± SE, t-value and * indicates a statistically significant predictor in the model. Nighttime sleep efficiency predicted next day’s LPA and daytime LPA predicted that nighttime sleep efficiency.

## Data Availability

Data sharing is not applicable to this article.

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
