# Peer review of "Bidirectional Associations between Physical Activity and Sleep in Early-Elementary-Age Latino Children with Obesity"

_sports, 2021, doi:10.3390/sports9020026_

Round 1

Reviewer 1 Report

Review Manuscript ID: sports-1060198 titled “Bidirectional associations between physical activity and sleep in early elementary age Latino children with obesity” by Merrigan et al.

This study investigated physical activity and sleep patterns in overweight/obese low-income Latino children. Main finding was that light physical activity was related to sleep patterns.

This is a well-conducted study and the manuscript is well written and has a good structure. The manuscript contains relevant data. I have no major criticism only minor comments.

How accurate is self-reported “Latino” – please back it up with a reference if possible.  

This is a small data set and the findings are preliminary and this should be mentioned.

In the discussion segment concerning the percentage that fulfil the physical activity guidelines. You should address the fact that this is completely dependent on which accelerometer cut-off points you choose. You choose those from Puyau et al. 2002, whereas others have chosen other accelerometer cut-off points.

Reviewer 2 Report

Thank you very much for allowing me to review the article “short communication”: “Bidirectional associations between physical activity and sleep in early elementary age Latino children with obesity” (Sport-1060198).

The aim this study was to investigate the bidirectional associations among physical activity and sleep patterns in over-weight / obese Latino children of low SES.

 The factors that influence the prevalence of obesity in children, include culture, environment, behaviors, and low socioeconomic status (SES) are a highly topical topic. The approval of the study is presented to the University Institutional Review Board for Human Subjects (# 1094469). Please indicate the university.

Comments:

Introduction: I suggest that the hypothesis should be prior to the objective of the study.

Material and methods: The calculation of the studied sample size must be presented. It is a very small sample size, perhaps this is the main problem of this interesting work.

Table 1.- The z-score should be used for weight, height and BMI.

Discussion: It remains to assess the limitations of the study, especially in relation to the sample size and its effects on a possible type II error.

Reviewer 3 Report

Abstract

Background: Briefly explain why it is important to know the patterns of sleep and physical activity in this group of people. The fact that there are no studies on this is fine but not determinative.
Include specific results for all the indicated variables.
Rewrite the conclusion. What does bidirectional mean? I think it is necessary to explain it better.

Introduction

Very well directed. However, I miss some concrete data from studies that have talked about the two-way relationship between physical exercise and sleep. Likewise, explain the differences between the duration and quality of sleep that they can have with physical exercise.

Material and methods

Although the authors direct us to a previous study of their group, I believe it is important that the sample be clearly described and how it was obtained.

Results

Include R2 in Table 2 to see the degree of strength of these associations.
Taking into account that the results shown in table 2 are the important part of the study, these results should be deeply described in the results section. It is not clear if this two-way relationship is with both aspects of the dream or only one of them.

Discussion

I believe that the authors should delve deeper into the physiological aspects that could relate the efficiency and duration of sleep with physical activity. We must remind the reader that this two-way relationship from what I have understood is more related to efficiency than to sleep duration.

Include a section on limitations, strengths and future lines of research related to the study carried out.

Conclusion

Perhaps I have not understood the results well, but that two-way relationship that the authors indicate is only in the efficiency and not in the duration of sleep. So the results both in this section and in the Abstract should go along that line.

Reviewer 4 Report

I want to thank you for the opportunity to review the article entitled: Bidirectional associations between physical activity and sleep in early elementary age Latino children with obesity to Sports. Although a lot of studies have examined the relationship between physical activity (PA) and sleep patterns and different health indicators (Saunders et al., 2016), less studies have examined the bidirectional associations between PA and sleep patterns, using objective measures on both health-related behaviours (Lang et al., 2016) and, particularly, in early elementary age Latino children with obesity. Therefore, it has to be noted that this study makes an important contribution to the literature in this research field. The study seems well designed and well presented. It is a study of good quality that I think will be relevant for the readers of Sport. However, there are minor concerns that have to be taken into account to improve the quality of this manuscript. Abstract 1.- Authors said “Among Latino children who are overweight/obese, a bidirectional relationship between LPA and sleep existed, although further investigation is required to determine causality” I would be more cautious in this statement because of the type of design, the type of sample and the sample number. For example, These results suggest that… 2.- I would add in the keywords, exercise, youth, and body mass index. Introducction 1.- Please, review these references to strenght the introduction section. Many of the most important articles are missing from the introduction. The authors need to identify pertinent literature and explain how their study builds upon and extends such literatura. If there has already been a lot of research on the specific topic, the included studies should ideally come from samples or research participants that share similar characteristics to the manuscript’s sample. As a general rule of thumb, the richer the literature on a topic, the narrower the selection of pertinent studies should be, so that it is clear to the reader what gaps in knowledge are addressed in the manuscript. Poitras, V. J., Gray, C. E., Borghese, M. M., Carson, V., Chaput, J. P., Janssen, I., ... & Tremblay, M. S. (2016). Systematic review of the relationships between objectively measured physical activity and health indicators in school-aged children and youth. Applied Physiology, Nutrition, and Metabolism, 41(6), S197-S239. Chaput, J. P., Gray, C. E., Poitras, V. J., Carson, V., Gruber, R., Olds, T., ... & Tremblay, M. S. (2016). Systematic review of the relationships between sleep duration and health indicators in school-aged children and youth. Applied physiology, nutrition, and metabolism, 41(6), S266-S282. Ávila-García, M., Femia-Marzo, P., Huertas-Delgado, F. J., & Tercedor, P. (2020). Bidirectional associations between objective physical activity and sleep patterns in Spanish school children. International journal of environmental research and public health, 17(3), 710. Antczak, D., Sanders, T., del Pozo Cruz, B., Parker, P., & Lonsdale, C. (2020). Day-to-day and longer-term longitudinal associations between physical activity, sedentary behavior, and sleep in children. Sleep. Ohayon, M.; Wickwire, E.M.; Hirshkowitz, M.; Albert, S.M.; Avidan, A.; Daly, F.J.; Dauvilliers, Y.; Ferri, R.; Fung, C.; Gozal, D.; et al. National Sleep Foundation’s sleep quality recommendations: First report. Sleep Health 2017, 3, 6–19 Mcneil, J.; Tremblay, M.S.; Leduc, G.; Boyer, C.; Bélanger, P.; Leblanc, A.G.; Borghese, M.M.; Chaput, J.P. Objectively-measured sleep and its association with adiposity and physical activity in a sample of Canadian children. J. Sleep Res. 2015, 24, 131–139. [CrossRef] Vincent, G.E.; Barnett, L.M.; Lubans, D.R.; Salmon, J.; Timperio, A.; Ridgers, N.D. Temporal and bidirectional associations between physical activity and sleep in primary school-aged children. Appl. Physiol. Nutr. Metab. 2016, 42, 238–242. [CrossRef] Soric, M.; Starc, G.; Borer, K.T.; Jurak, G.; Kovaˇc, M.; Strel, J.; Mišigoj-Durakovi´c, M. Associations of objectively assessed sleep and physical activity in 11-year old children. Ann. Hum. Biol. 2015, 42, 31–37. [CrossRef] Ekstedt, M.; Nyberg, G.; Ingre, M.; Ekblom, O.; Marcus, C. Sleep, physical activity and BMI in six to ten-year-old children measured by accelerometry: A cross-sectional study. Int. J. Behav. Nutr. Phys. Act. 2013, 10, 82. [CrossRef] Lin, Y.; Borghese, M.M.; Janssen, I. Bi-directional association between sleep and outdoor active play among 10–13 year olds. BMC Public Health 2018, 18, 224. [CrossRef] [PubMed] Lang, C., Kalak, N., Brand, S., Holsboer-Trachsler, E., Pühse, U., & Gerber, M. (2016). The relationship between physical activity and sleep from mid adolescence to early adulthood. A systematic review of methodological approaches and meta-analysis. Sleep medicine reviews, 28, 32-45. 2.- Author said “However, younger children (6-11 years) appear to meet or exceed physical activity guidelines of at least 60 minutes of moderate to vigorous physical activity (MVPA), regardless of ethnicity and weight.[3,5]” Meet or exceed? Please, review these references to strenght this sentence. Hnatiuk, J. A., Salmon, J., Hinkley, T., Okely, A. D., & Trost, S. (2014). A review of preschool children’s physical activity and sedentary time using objective measures. American Journal of Preventive Medicine, 47(4), 487-497. 3.- Authors said “Nightly sleep duration ranging from 9-12 hours has been suggested for children aged 6-12 years.[7]” However, these recommendations are not in line with those proposed by the following references. Hirshkowitz, M.; Whiton, K.; Albert, S.M.; Alessi, C.; Bruni, O.; DonCarlos, L.; Hazen, N.; Herman, J.; Katz, E.S.; Kheirandish-Gozal, L.; et al. National sleep foundation’s sleep time duration recommendations: Methodology and results summary. Sleep Health 2015, 1, 40–43. Tremblay, M. S., Carson, V., Chaput, J. P., Connor Gorber, S., Dinh, T., Duggan, M., ... & Zehr, L. (2016). Canadian 24-hour movement guidelines for children and youth: an integration of physical activity, sedentary behaviour, and sleep. Applied Physiology, Nutrition, and Metabolism, 41(6), S311-S327. 4.- Authors said “Therefore, associations between sleep and physical activity are likely 52 bidirectional, but research is lacking.[11–13]” I would add that there is a lack of studies in the specific population of your work (weight/obese children of low SES). 5.- Authors said “It was hypothesized that overweight/obese 59 children with more MVPA would also have longer sleep durations and improved sleep 60 quality that night; and likewise, a night with higher quantity and quality of sleep would 61 increase MVPA the next day.” The hypotheses have to be supported by previous studies. I am not sure that there are studies in this particular population to establish hypotheses in this study. Material and methods 1.- Please, remove this sentence in the Design and Participants section. “The present study examined bidirectional associations of objectively measured physical 67 activity and sleep, by first examining daytime physical activity in relation to that night’s 68 sleep, and then nighttime sleep in relation to the next day’s physical activity” The objective does not need to be highlighted here. 2.- What does mean self-reported Latino? 3.- Please, put this sentene in the results section “Physical characteristics, as well as physical activity and sleep data, are provided in Table 76 1. Data collection procedures for anthropometrics and body composition are described 77 elsewhere.[6]” 4.- Please, add the type of design and the sample of study in the participants section. Measures 1.- Why did you use 60 epoch in young children? Please, see Aibar, A., & Chanal, J. (2015). Physical education: the effect of epoch lengths on children’s physical activity in a structured context. PloS one, 10(4), e0121238. Aibar, A., Bois, J. E., Zaragoza, J., Generelo, E., Julián, J. A., & Paillard, T. (2014). Do epoch lengths affect adolescents' compliance with physical activity guidelines?. The Journal of sports medicine and physical fitness, 54(3), 326-334. Migueles, J. H., Cadenas-Sanchez, C., Ekelund, U., Nyström, C. D., Mora-Gonzalez, J., Löf, M., ... & Ortega, F. B. (2017). Accelerometer data collection and processing criteria to assess physical activity and other outcomes: a systematic review and practical considerations. Sports medicine, 47(9), 1821-1845. 2.- Why were the two behaviours only measured for 6 days? Usually at least 7 days are measured. 3.- Authors said “Participants that 88 removed their device for extended periods were removed from analysis (n=5)” What does mean extended periods? Please add the number of hours the accelerometers were taken to be valid on that day. Please add the number of valid days during the week and the weekend that were necessary for the measure to be valid. Discussion 1.- Authors said “This study investigated bidirectional associations between physical activity and sleep patterns in young Latino children of low SES with obesity” However, in the introduction authors said “…in overweight/obese” Please, add …SES with overweight or obesity. This aspect should be corrected in the discussion and conclusions. 2.- Authors said “One possible explanation is the inherently greater physical activity levels of children, during the day and night, that would reduce the amount of time left in the day to sleep”. Please, supports this sentence with the 24-hour paradigm. Tremblay, M. S., Carson, V., Chaput, J. P., Connor Gorber, S., Dinh, T., Duggan, M., ... & Zehr, L. (2016). Canadian 24-hour movement guidelines for children and youth: an integration of physical activity, sedentary behaviour, and sleep. Applied Physiology, Nutrition, and Metabolism, 41(6), S311-S327. Chaput, J. P., Carson, V., Gray, C. E., & Tremblay, M. S. (2014). Importance of all movement behaviors in a 24 hour period for overall health. International journal of environmental research and public health, 11(12), 12575-12581. 3.- Please, review the references mentioned in the introduction to strenght the introduction section. 4.- After discussing all their findings, authors should summarize the major strengths and limitations of their work, with reference to (where relevant) conceptual development or refinement, methodology, and relevance for practice and policy. Authors should be honest about the limitations of their work; some well-known limitations do not need detailed discussion. For instance, most readers are familiar with the limitations of cross-sectional designs, hence, if word count is tight, authors could simply mention the cross-sectional design as being a limitation of their work, without delving further into this, or they can provide a reference to work where such limitations are discussed in detail.

Round 2

Reviewer 2 Report

Thank you very much for allowing me to review the short communication again. “short communication”: “Bidirectional associations between physical activity and sleep in early elementary age Latino children with obesity” (Sport-1060198).

 This is an interesting work its aims to investigate the bidirectional associations among physical activity and sleep patterns in overweight / obese Latino children of low SES.

Both in the letter from the authors to the comments made and the new document, they adequately respond to the suggestions made.

I fully agree with the suggestions made by the other reviewers, as well, which I think have improved the quality of the work

.

However, it is a work with a very small sample size, this size is not justified with the objectives of the study, these aspects are valued by the authors in the discussion.

 Table 2 is really the most important part of the article and should be better explained.

Author Response

Thank you for your comments. Since the results in Table 2 have been expanded on in the results section and throughout the discussion, we felt as though the confusion of its reporting may have been a result of the Table's layout. We have adjusted the table layout and have added to the endnote further explanation of what the significant predictors were. The meaning behind the estimate values were explained in the results section Lines 150-153.

Reviewer 3 Report

The authors have improved the manuscript. I think now it is ok.

Author Response

Thank you.